# Real-Time Reinforcement Learning

**Simon Ramstedt**
Mila, Element AI,
Université de Montréal
simonramstedt@gmail.com

**Christopher Pal**
Mila, Element AI,
Polytechnique Montréal
christopher.pal@polymtl.ca

## Abstract

Markov Decision Processes (MDPs), the mathematical framework underlying most algorithms in Reinforcement Learning (RL), are often used in a way that wrongfully assumes that the state of an agent's environment does not change during action selection. As RL systems based on MDPs begin to find application in real-world, safety-critical situations, this mismatch between the assumptions underlying classical MDPs and the reality of real-time computation may lead to undesirable outcomes. In this paper, we introduce a new framework, in which states and actions evolve simultaneously and show how it is related to the classical MDP formulation. We analyze existing algorithms under the new real-time formulation and show why they are suboptimal when used in real time. We then use those insights to create a new algorithm Real-Time Actor-Critic (RTAC) that outperforms the existing state-of-the-art continuous control algorithm Soft Actor-Critic both in real-time and non-real-time settings. Code and videos can be found at `github.com/rmst/rtrl`.

Reinforcement Learning, has led to great successes in games (Tesauro, 1994; Mnih et al., 2015; Silver et al., 2017) and is starting to be applied successfully to real-world robotic control (Schulman et al., 2015; Hwangbo et al., 2019).

The theoretical underpinning for most methods in Reinforcement Learning is the Markov Decision Process (MDP) framework (Bellman, 1957). While it is well suited to describe turn-based decision problems such as board games, this framework is ill suited for real-time applications in which the environment's state continues to evolve while the agent selects an action (Travnik et al., 2018). Nevertheless, this framework has been used for real-time problems using what are essentially tricks, e.g. pausing a simulated environment during action selection or ensuring that the time required for action selection is negligible (Hwangbo et al., 2017).

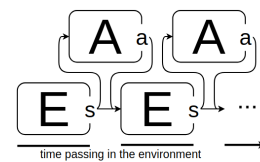

Figure 1: Turn-based interaction

Instead of relying on such tricks, we propose an augmented decision-making framework - Real-Time Reinforcement Learning (RTRL) - in which the agent is allowed exactly one timestep to select an action. RTRL is conceptually simple and opens up new algorithmic possibilities because of its special structure.

We leverage RTRL to create Real-Time Actor-Critic (RTAC), a new actor-critic algorithm, better suited for real-time interaction, that is based on Soft Actor-Critic (Haarnoja et al., 2018a). We then show experimentally that RTAC outperforms SAC in both real-time and non-real-time settings.

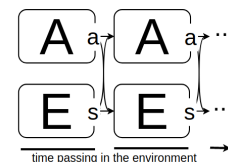

Figure 2: Real-time interaction

# 1 Background

In Reinforcement Learning the world is split up into agent and environment. The agent is represented by a policy – a state-conditioned action distribution, while the environment is represented by a Markov Decision Process (Def. 1). Traditionally, the agent-environment interaction has been governed by the MDP framework. Here, however, we strictly use MDPs to represent the environment. The agent-environment interaction is instead described by different types of Markov Reward Processes (MRP), with the *TBMRP* (Def. 2) behaving like the traditional interaction scheme.

**Definition 1.** *A Markov Decision Process (MDP) is characterized by a tuple with*

*(1) state space $S$,   (2) action space $A$,   (3) initial state distribution $\mu : S \to \mathbb{R}$,*
*(4) transition distribution $p : S \times S \times A \to \mathbb{R}$,   (5) reward function $r : S \times A \to \mathbb{R}$.*

An agent-environment system can be condensed into a Markov Reward Process $(S, \mu, \kappa, \bar{r})$ consisting of a Markov process $(S, \mu, \kappa)$ and a state-reward function $\bar{r}$. The Markov process induces a sequence of states $(s_t)_{t \in \mathbb{N}}$ and, together with $\bar{r}$, a sequence of rewards $(r_t)_{t \in \mathbb{N}} = (\bar{r}(s_t))_{t \in \mathbb{N}}$.

As usual, the objective is to find a policy that maximizes the expected sum of rewards. In practice, rewards can be discounted and augmented to guarantee convergence, reduce variance and encourage exploration. However, when evaluating the performance of an agent, we will always use the undiscounted sum of rewards.

## 1.1 Turn-Based Reinforcement Learning

Usually considered part of the standard Reinforcement Learning framework is the turn-based scheme in which agent and environment interact. We call this interaction scheme Turn-Based Markov Reward Process.

**Definition 2.** *A Turn-Based Markov Reward Process $(S, \mu, \kappa, \bar{r}) = TBMRP(E, \pi)$ combines a Markov Decision Process $E = (S, A, \mu, p, r)$ with a policy $\pi$, such that*

$$\kappa(s_{t+1}|s_t) = \int_A p(s_{t+1}|s_t, a)\pi(a|s_t)\,da \quad and \quad \bar{r}(s_t) = \int_A r(s_t, a)\pi(a|s_t)\,da. \quad (1)$$

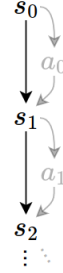

Figure 3: *TBMRP*

We say the interaction is turn-based, because the environment pauses while the agent selects an action and the agent pauses until it receives a new observation from the environment. This is illustrated in Figure 1. An action selected in a certain state is paired up again with that same state to induce the next. The state does not change during the action selection process.

# 2 Real-Time Reinforcement Learning

In contrast to the conventional, turn-based interaction scheme, we propose an alternative, real-time interaction framework in which states and actions evolve simultaneously. Here, agent and environment step in unison to produce new state-action pairs $\boldsymbol{x}_{t+1} = (s_{t+1}, a_{t+1})$ from old state-action pairs $\boldsymbol{x}_t = (s_t, a_t)$ as illustrated in Figures 2 and 4.

**Definition 3.** *A Real-Time Markov Reward Process $(\boldsymbol{X}, \boldsymbol{\mu}, \boldsymbol{\kappa}, \bar{\boldsymbol{r}}) = RTMRP(E, \boldsymbol{\pi})$ combines a Markov Decision Process $E = (S, A, \mu, p, r)$ with a policy $\pi$, such that*

$$\boldsymbol{\kappa}(s_{t+1}, a_{t+1}|s_t, a_t) = p(s_{t+1}|s_t, a_t)\,\boldsymbol{\pi}(a_{t+1}|s_t, a_t) \quad and \quad \bar{\boldsymbol{r}}(s_t, a_t) = r(s_t, a_t). \quad (2)$$

*The system state space is $\boldsymbol{X} = S \times A$. The initial action $a_0$ can be set to some fixed value, i.e. $\boldsymbol{\mu}(s_0, a_0) = \mu(s_0)\,\delta(a_0 - c)$.*[1]

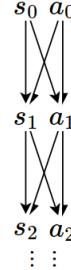

Figure 4: *RTMRP*

Note that we introduced a new policy $\boldsymbol{\pi}$ that takes state-action pairs instead of just states. That is because the system state $\boldsymbol{x} = (s, a)$ is now a state-action pair and $s$ alone is not a sufficient statistic of the future of the stochastic process anymore.

## 2.1 The real-time framework is made for back-to-back action selection

In the real-time framework, the agent has exactly one timestep to select an action. If an agent takes longer that its policy would have to be broken up into stages that take less than one timestep to evaluate. On the other hand, if an agent takes less than one timestep to select an action, the real-time framework will delay applying the action until the next observation is made. The optimal case is when an agent, immediately upon finishing selecting an action, observes the next state and starts computing the next action. This continuous, back-to-back action selection is ideal in that it allows the agent to update its actions the quickest and no delay is introduced through the real-time framework.

To achieve back-to-back action selection, it might be necessary to match timestep size to the policy evaluation time. With current algorithms, reducing timestep size might lead to worse performance. Recently, however, progress has been made towards timestep agnostic methods (Tallec et al., 2019). We believe back-to-back action selection is an achievable goal and we demonstrate here that the real-time framework is effective even if we are not able to tune timestep size (Section 5).

## 2.2 Real-time interaction can be expressed within the turn-based framework

It is possible to express real-time interaction within the standard, turn-based framework, which allows us to reconnect the real-time framework to the vast body of work in RL. Specifically, we are trying to find an augmented environment $RTMDP(E)$ that behaves the same with turn-based interaction as would $E$ with real-time interaction.

In the real-time framework the agent communicates its action to the environment via the state. However, in the traditional, turn-based framework, only the environment can directly influence the state. We therefore need to deterministically "pass through" the action to the next state by augmenting the transition function. The $RTMDP$ has two types of actions, (1) the actions $\boldsymbol{a}_t$ emitted by the policy and (2) the action component $a_t$ of the state $\boldsymbol{x}_t = (s_t, a_t)$, where $a_t = \boldsymbol{a}_{t-1}$ with probability one.

**Definition 4.** *A Real-Time Markov Decision Process* $(\boldsymbol{X}, A, \boldsymbol{\mu}, \boldsymbol{p}, \boldsymbol{r}) = RTMDP(E)$ *augments another Markov Decision Process* $E = (S, A, \mu, p, r)$*, such that*

*(1) state space* $\boldsymbol{X} = S \times A$,   *(2) action space is* $A$,
*(3) initial state distribution* $\boldsymbol{\mu}(\boldsymbol{x}_0) = \boldsymbol{\mu}(s_0, a_0) = \mu(s_0)\,\delta(a_0 - c)$,
*(4) transition distribution* $\boldsymbol{p}(\boldsymbol{x}_{t+1}|\boldsymbol{x}_t, \boldsymbol{a}_t) = \boldsymbol{p}(s_{t+1}, a_{t+1}|s_t, a_t, \boldsymbol{a}_t) = p(s_{t+1}|s_t, a_t)\,\delta(a_{t+1} - \boldsymbol{a}_t)$
*(5) reward function* $\boldsymbol{r}(\boldsymbol{x}_t, \boldsymbol{a}_t) = \boldsymbol{r}(s_t, a_t, \boldsymbol{a}_t) = r(s_t, a_t)$. <span style="font-size:smaller">*(tap to see code)*</span>

**Theorem 1.** [2] *A policy* $\boldsymbol{\pi} : A \times \boldsymbol{X} \to \mathbb{R}$ *interacting with* $RTMDP(E)$ *in the conventional, turn-based manner gives rise to the same Markov Reward Process as* $\boldsymbol{\pi}$ *interacting with* $E$ *in real-time, i.e.*

$$RTMRP(E, \boldsymbol{\pi}) = TBMRP(RTMDP(E), \boldsymbol{\pi}). \tag{3}$$

Interestingly, the RTMDP is equivalent to a 1-step constant delay MDP (Walsh et al. (2008)). However, we believe the different intuitions behind both of them warrant the different names: The constant delay MDP is trying to model external action and observation delays whereas the RTMDP is modelling the time it takes to select an action. The connection makes sense, though: In a framework where the action selection is assumed to be instantaneous, we can apply a delay to account for the fact that the action selection was not instantaneous after all.

## 2.3 Turn-based interaction can be expressed within the real-time framework

It is also possible to define an augmentation $TBMDP(E)$ that allows us to express turn-based environments (e.g. Chess, Go) within the real-time framework (Def. 7 in the Appendix). By assigning separate timesteps to agent and environment, we can allow the agent to act while the environment pauses. More specifically, we add a binary variable $b$ to the state to keep track of whether it is the environment's or the agent's turn. While $b$ inverts at every timestep, the underlying environment only advances every other timestep.

**Theorem 2.** *A policy* $\boldsymbol{\pi}(\boldsymbol{a}|s, b, a) = \pi(\boldsymbol{a}|s)$ *interacting with* $TBMDP(E)$ *in real time, gives rise to a Markov Reward Process that contains (Def. 10) the MRP resulting from* $\pi$ *interacting with* $E$ *in the conventional, turn-based manner, i.e.*

$$TBMRP(E, \pi) \propto RTMRP(TBMDP(E), \boldsymbol{\pi}) \tag{4}$$

As a result, not only can we use conventional algorithms in the real-time framework but we can use algorithms built on the real-time framework for all turn-based problems.

# 3 Reinforcement Learning in Real-Time Markov Decision Processes

Having established the RTMDP as a compatibility layer between conventional RL and RTRL, we can now look how existing theory changes when moving from an environment $E$ to $RTMDP(E)$.

Since most RL methods assume that the environment's dynamics are completely unknown, they will not be able to make use of the fact that we precisely know part of the dynamics of RTMDP. Specifically they will have to learn from data, the effects of the "feed-through" mechanism which could lead to much slower learning and worse performance when applied to an environment $RTMDP(E)$ instead of $E$. This could especially hurt the performance of off-policy algorithms which have been among the most successful RL methods to date (Mnih et al., 2015; Haarnoja et al., 2018a). Most off-policy methods make use of the action-value function.

**Definition 5.** *The action value function $q_E^\pi$ for an environment $E = (S, A, \mu, p, r)$ and a policy $\pi$ can be recursively defined as*

$$q_E^\pi(s_t, a_t) = r(s_t, a_t) + \mathbb{E}_{s_{t+1} \sim p(\cdot|s_t, a_t)}[\mathbb{E}_{a_{t+1} \sim \pi(\cdot|s_{t+1})}[q_E^\pi(s_{t+1}, a_{t+1})]] \quad (5)$$

When this identity is used to train an action-value estimator, the transition $s_t, a_t, s_{t+1}$ can be sampled from a replay memory containing off-policy experience while the next action $a_{t+1}$ is sampled from the policy $\pi$.

**Lemma 1.** *In a Real-Time Markov Decision Process for the action-value function we have*

$$q_{RTMDP(E)}^{\boldsymbol{\pi}}(\underline{s_t, a_t}, \boldsymbol{a}_t) = r(s_t, a_t) + \mathbb{E}_{s_{t+1} \sim p(\cdot|s_t, a_t)}[\mathbb{E}_{\boldsymbol{a}_{t+1} \sim \boldsymbol{\pi}(\cdot|\underline{s_{t+1}, \boldsymbol{a}_t})}[q_{RTMDP(E)}^{\boldsymbol{\pi}}(\underline{s_{t+1}, \boldsymbol{a}_t}, \boldsymbol{a}_{t+1})]] \quad (6)$$

Note that the action $\boldsymbol{a}_t$ does not affect the reward nor the next state. The only thing that $\boldsymbol{a}_t$ does affect is $a_{t+1}$ which, in turn, only in the next timestep will affect $r(s_{t+1}, a_{t+1})$ and $s_{t+2}$. To learn the effect of an action on $E$ (specifically the future rewards), we now have to perform two updates where previously we only had to perform one. We will investigate experimentally the effect of this on the off-policy Soft Actor-Critic algorithm (Haarnoja et al., 2018a) in Section 5.1.

## 3.1 Learning the state-value function off-policy

The state-value function can usually not be used in the same way as the action-value function for off-policy learning.

**Definition 6.** *The state-value function $v_E^\pi$ for an environment $E = (S, A, \mu, p, r)$ and a policy $\pi$ is*

$$v_E^\pi(s_t) = \mathbb{E}_{a_t \sim \pi(\cdot|s_t)}[r(s_t, a_t) + \mathbb{E}_{s_{t+1} \sim p(\cdot|s_t, a_t)}[v_E^\pi(s_{t+1})]] \quad (7)$$

The definition shows that the expectation over the action is taken *before* the expectation over the next state. When using this identity to train a state-value estimator, we cannot simply change the action distribution to allow for off-policy learning since we have no way of resampling the next state.

**Lemma 2.** *In a Real-Time Markov Decision Process for the state-value function we have*

$$v_{RTMDP(E)}^{\boldsymbol{\pi}}(\underline{s_t}, a_t) = r(s_t, a_t) + \mathbb{E}_{s_{t+1} \sim p(\cdot|s_t, a_t)}[\mathbb{E}_{\boldsymbol{a}_t \sim \boldsymbol{\pi}(\cdot|\underline{s_t}, a_t)}[v_{RTMDP(E)}^{\boldsymbol{\pi}}(\underline{s_{t+1}, \boldsymbol{a}_t})]]. \quad (8)$$

Here, $s_t, a_t, s_{t+1}$ is always a valid transition no matter what action $\boldsymbol{a}_t$ is selected. Therefore, when using the real-time framework, we can use the value function for off-policy learning. Since Equation 8 is the same as Equation 5 (except for the policy inputs), we can use the state-value function where previously the action-value function was used without having to learn the dynamics of the $RTMDP$ from data since they have already been applied to Equation 8.

## 3.2 Partial simulation

The off-policy learning procedure described in the previous section can be applied more generally. Whenever parts of the agent-environment system are known and (temporarily) independent of the remaining system, they can be used to generate synthetic experience. More precisely, transitions with a start state $s = (w, z)$ can be generated according to the true transition kernel $\kappa(s'|s)$ by simulating the known part of the transition $(w \to w')$ and using a stored sample for the unknown part of the transition $(z \to z')$. This is only possible if the transition kernel factorizes as $\kappa(w', z'|s) = \kappa_{\text{known}}(w'|s) \, \kappa_{\text{unknown}}(z'|s)$. Hindsight Experience Replay (Andrychowicz et al., 2017) can be seen as another example of partial simulation. There, the goal part of the state evolves independently of the rest which allows for changing the goal in hindsight. In the next section, we use the same partial simulation principle to compute the gradient of the policy loss.

# 4 Real-Time Actor-Critic (RTAC)

Actor-Critic algorithms (Konda & Tsitsiklis, 2000) formulate the RL problem as bi-level optimization where the critic evaluates the actor as accurately as possible while the actor tries to improve its evaluation by the critic. Silver et al. (2014) showed that it is possible to reparameterize the actor evaluation and directly compute the pathwise derivative from the critic with respect to the actor parameters and thus telling the actor how to improve. Heess et al. (2015) extended that to stochastic policies and Haarnoja et al. (2018a) further extended it to the maximum entropy objective to create Soft Actor-Critic (SAC) which RTAC is going to be based on and compared against.

In SAC a parameterized policy $\pi$ (the actor) is optimized to minimize the KL-divergence between itself and the exponential of an approximation of the action-value function $q$ (the critic) normalized by $Z$ (where $Z$ is unknown but irrelevant to the gradient) giving rise to the policy loss

$$L_{E,\pi}^{\text{SAC}} = \mathbb{E}_{s_t \sim D} D_{\text{KL}}(\pi(\cdot|s_t)|| \exp(\tfrac{1}{\alpha}q(s_t,\cdot))/Z(s_t))^3 \tag{9}$$

where $D$ is a uniform distribution over the replay memory containing past states, actions and rewards. The action-value function itself is optimized to fit Equation 5 presented in the previous section (augmented with an entropy term). We can thus expect SAC to perform worse in RTMDPs.

In order to create an algorithm better suited for the real-time setting we propose to use a state-value function approximator $v$ as the critic instead, that will give rise to the same policy gradient.

**Proposition 1.** *The following policy loss based on the state-value function*

$$L_{RTMDP(E),\pi}^{RTAC} = \mathbb{E}_{(s_t,a_t)\sim D}\mathbb{E}_{s_{t+1}\sim p(\cdot|s_t,a_t)} D_{KL}(\pi(\cdot|s_t,a_t)|| \exp(\tfrac{1}{\alpha}\gamma v(s_{t+1},\cdot))/Z(s_{t+1})) \tag{10}$$

*has the same policy gradient as $L_{RTMDP(E),\pi}^{SAC}$, i.e.*

$$\nabla_{\pi} L_{RTMDP(E),\pi}^{RTAC} = \nabla_{\pi} L_{RTMDP(E),\pi}^{SAC} \tag{11}$$

The value function itself is trained off-policy according to the procedure described in Section 3.1 to fit an augmented version of Equation 8, specifically

$$v_{\text{target}} = r(s_t,a_t) + \mathbb{E}_{s_{t+1}\sim p(\cdot|s_t,a_t)}[\mathbb{E}_{a_t \sim \pi(\cdot|s_t,a_t)}[\bar{v}_{\bar{\theta}}((s_{t+1},a_t)) - \alpha \log(\pi(a_t|s_t,a_t))]]. \tag{12}$$

Therefore, for the value loss, we have

$$L_{RTMDP(E),v}^{\text{RTAC}} = \mathbb{E}_{(x_t,r_t,s_{t+1})\sim D}[(v(x_t) - v_{\text{target}})^2] \tag{13}$$

## 4.1 Merging actor and critic

Using the state-value function as the critic has another advantage: When evaluated at the same timestep, the critic does not depend on the actor's output anymore and we are therefore able to use a single neural network to represent both the actor and the critic. Merging actor and critic makes it necessary to trade off between the value function and policy loss. Therefore, we introduce an additional hyperparameter $\beta$.

$$L(\theta) = \beta L_{RTMDP(E),\pi_\theta}^{\text{RTAC}} + (1-\beta)L_{RTMDP(E),v_\theta}^{\text{RTAC}} \tag{14}$$

Merging actor and critic could speed up learning and even improve generalization, but could also lead to greater instability. We compare RTAC with both merged and separate actor and critic networks in Section 5.

## 4.2 Stabilizing learning

Actor-Critic algorithms are known to be unstable during training. We use a number of techniques that help make training more stable. Most notably we use Pop-Art output normalization (van Hasselt et al., 2016) to normalize the value targets. This is necessary if $v$ and $\pi$ are represented using an overlapping set of parameters. Since the scale of the error gradients of the value loss is highly non-stationary it

is hard to find a good trade-off between policy and value loss ($\beta$). If $v$ and $\pi$ are separate, Pop-Art matters less, but still improves performance both in SAC as well as in RTAC.

Another difficulty are the recursive value function targets. Since we try to maximize the value function, overestimation errors in the value function approximator are amplified and recursively used as target values in the following optimization steps. As introduced by Fujimoto et al. (2018) and like SAC, we will use two value function approximators and take their minimum when computing the target values to reduce value overestimation, i.e. $\bar{\boldsymbol{v}}_{\bar{\theta}}(\cdot) = \min_{i \in \{1,2\}} \boldsymbol{v}_{\bar{\theta},i}(\cdot)$.

Lastly, to further stabilize the recursive value function estimation, we use target networks that slowly track the weights of the network (Mnih et al., 2015; Lillicrap et al., 2015), i.e. $\bar{\theta} \leftarrow \tau\theta + (1-\tau)\bar{\theta}$. The tracking weights $\bar{\theta}$ are then used to compute $\boldsymbol{v}_{\text{target}}$ in Equation 12.

## 5   Experiments

We compare Real-Time Actor-Critic to Soft Actor-Critic (Haarnoja et al., 2018a) on several OpenAI-Gym/MuJoCo benchmark environments (Brockman et al., 2016; Todorov et al., 2012) as well as on two Avenue autonomous driving environments with visual observations (Ibrahim et al., 2019).

The SAC agents used for the results here, include both an action-value and a state-value function approximator and use a fixed entropy scale $\alpha$ (as in Haarnoja et al. (2018a)). In the code accompanying this paper we dropped the state-value function approximator since it had no impact on the results (as done and observed in Haarnoja et al. (2018b)). For a comparison to other algorithms such as DDPG, PPO and TD3 also see Haarnoja et al. (2018a,b).

To make the comparison between the two algorithms as fair as possible, we also use output normalization in SAC which improves performance on all tasks (see Figure 9 in Appendix A for a comparison between normalized and unnormalized SAC). Both SAC and RTAC are performing a single optimization step at every timestep in the environment starting after the first 10000 timesteps of collecting experience based on the initial random policy. The hyperparameters used can be found in Table 1.

### 5.1   SAC in Real-Time Markov Decision Processes

When comparing the return trends of SAC in turn-based environments $E$ against SAC in real-time environments $RTMDP(E)$, the performance of SAC deteriorates. This seems to confirm our hypothesis that having to learn the dynamics of the augmented environment from data impedes action-value function approximation (as hypothesized in Section 3).

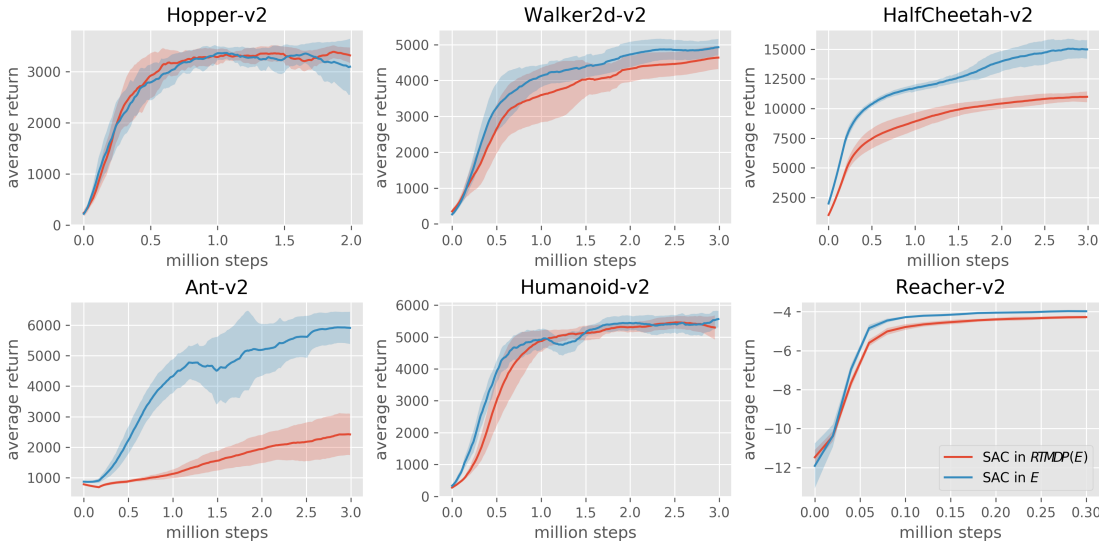

Figure 5: Return trends for SAC in turn-based environments $E$ and real-time environments $RTMDP(E)$. Mean and 95% confidence interval are computed over eight training runs per environment.

## 5.2 RTAC and SAC on MuJoCo in real time

Figure 6 shows a comparison between RTAC and SAC in real-time versions of the benchmark environments. We can see that RTAC learns much faster and achieves higher returns than SAC in $RTMDP(E)$. This makes sense as it does not have to learn from data the "pass-through" behavior of the RTMDP. We show RTAC with separate neural networks for the policy and value components showing that a big part of RTAC's advantage over SAC is its value function update. However, the fact that policy and value function networks can be merged further improves RTAC's performance as the plots suggest. Note that RTAC is always in $RTMDP(E)$, therefore we do not explicitly state it again.

RTAC is even outperforming SAC in $E$ (when SAC is allowed to act without real-time constraints) in four out of six environments including the two hardest ones - Ant and Humanoid - with largest state and action space (Figure 11). We theorize this is possible due to the merged actor and critic networks used in RTAC. It is important to note however, that for RTAC with merged actor and critic networks output normalization is critical (Figure 12).

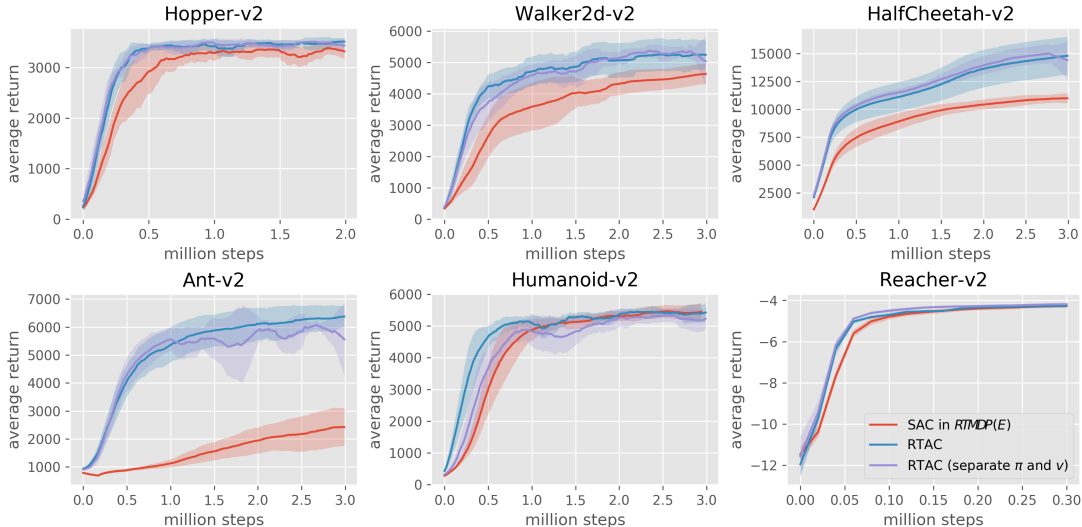

Figure 6: Comparison between RTAC and SAC in RTMDP versions of the benchmark environments. Mean and $95\%$ confidence interval are computed over eight training runs per environment.

## 5.3 Autonomous driving task

In addition to the MuJoCo environments, we have also tested RTAC and SAC on an autonomous driving task using the Avenue simulator (Ibrahim et al., 2019). Avenue is a game-engine-based simulator where the agent controls a car. In the task shown here, the agent has to stay on the road and possibly steer around pedestrians. The observations are single image (256x64 grayscale pixels) and the car's velocity. The actions are continuous and two dimensional, representing steering angle and gas-brake. The agent is rewarded proportionally to the car's velocity in the direction of the road and negatively rewarded when making contact with a pedestrian or another car. In addition, episodes are terminated when leaving the road or colliding with any objects or pedestrians.

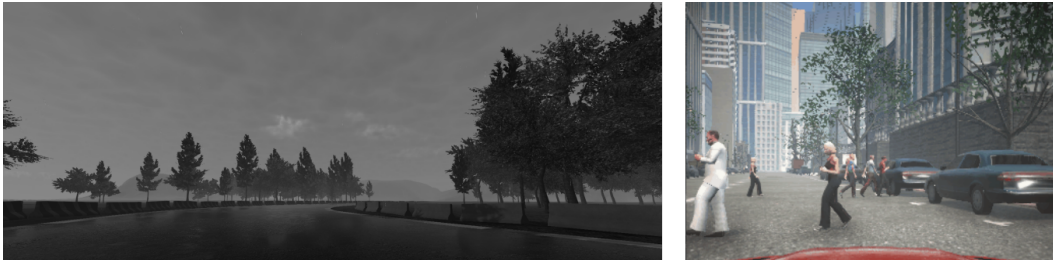

Figure 7: Left: Agent's view in `RaceSolo`. Right: Passenger view in `CityPedestrians`.

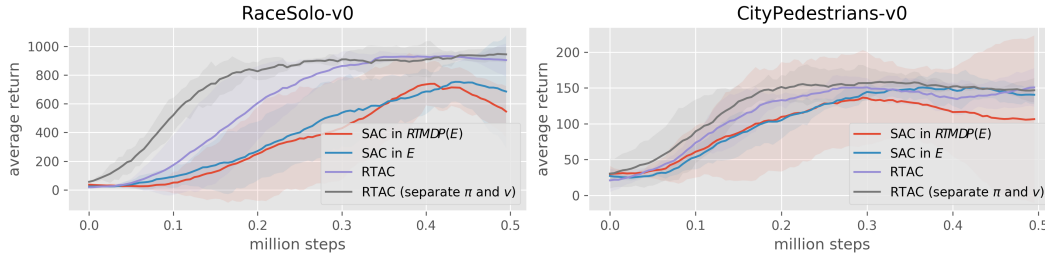

Figure 8: Comparison between RTAC and SAC in RTMDP versions of the autonomous driving tasks. We can see that RTAC under real-time constraints outperforms SAC even without real-time constraints. Mean and 95% confidence interval are computed over four training runs per environment.

The hyperparameters used for the autonomous driving task are largely the same as for the MuJoCo tasks, however we used a lower entropy reward scale (0.05) and lower learning rate (0.0002). We used convolutional neural networks with four layers of convolutions with filter sizes $(8, 4, 4, 4)$, strides $(2, 2, 2, 1)$ and $(64, 64, 128, 128)$ channels. The convolutional layers are followed by two fully connected layers with $512$ units each.

## 6 Related work

Travnik et al. (2018) noticed that the traditional MDP framework is ill suited for real-time problems. Other than our paper, however, no rigorous framework is proposed as an alternative, nor is any theoretical analysis provided.

Firoiu et al. (2018) applies a multi-step action delay to level the playing field between humans and artificial agents on the ALE (Atari) benchmark However, it does not address the problems arising from the turn-based MDP framework or recognizes the significance and consequences of the one-step action delay.

Similar to RTAC, NAF (Gu et al., 2016) is able to do continuous control with a single neural network. However, it is requiring the action-value function to be quadratic in the action (and thus possible to optimize in closed form). This assumption is quite restrictive and could not outperform more general methods such as DDPG.

In SVG(1) (Heess et al., 2015) a differentiable transition model is used to compute the path-wise derivative of the value function one timestep after the action selection. This is similar to what RTAC is doing when using the value function to compute the policy gradient. However, in RTAC, we use the actual differentiable dynamics of the RTMDP, i.e. "passing through" the action to the next state, and therefore we do not need to approximate the transition dynamics. At the same time, transitions for the underlying environment are not modelled at all and instead sampled which is only possible because the actions $\boldsymbol{a}_t$ in a RTMDP only start to influence the underlying environment at the next timestep.

## 7 Discussion

We have introduced a new framework for Reinforcement Learning, RTRL, in which agent and environment step in unison to create a sequence of state-action pairs. We connected RTRL to the conventional Reinforcement Learning framework through the RTMDP and investigated its effects in theory and practice. We predicted and confirmed experimentally that conventional off-policy algorithms would perform worse in real-time environments and then proposed a new actor-critic algorithm, RTAC, that not only avoids the problems of conventional off-policy methods with real-time interaction but also allows us to merge actor and critic which comes with an additional gain in performance. We showed that RTAC outperforms SAC on both a standard, low dimensional continuous control benchmark, as well as a high dimensional autonomous driving task.

**Acknowledgments**

We would like to thank Cyril Ibrahim for building and helping us with the Avenue simulator; Craig Quiter and Sherjil Ozair for insightful discussions about agent-environment interaction; Alex Piché, Scott Fujimoto, Bhairav Metha and Jhelum Chakravorty, for reading drafts of this paper and finally Jose Gallego, Olexa Bilaniuk and many others at Mila that helped us on countless occasions online and offline.

This work was completed during a part-time internship at Element AI and was supported by the Open Philanthropy Project.

## Footnotes

[1]$\delta$ is the Dirac delta distribution. If $y \sim \delta(\cdot - x)$ then $y = x$ with probability one.

[2]All proofs are in Appendix C.

[3]$\alpha$ is a temperature hyperparameter. For $\alpha \to 0$, the maximum entropy objective reduces to the traditional objective. To compare with the hyperparameters table we have $\alpha = \frac{\text{entropy scale}}{\text{reward scale}}$.

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
