[Supplementary Material]

# A    Additional Experiments

Figure 9: SAC with and without output normalization. SAC in $E$ (no output norm) corresponds to the canonical version presented in Haarnoja et al. (2018a). Mean and 95% confidence interval are computed over eight training runs per environment.

Figure 10: Comparison between different actor loss scales ($\beta$). Mean and 95% confidence interval are computed over four training runs per environment.

Figure 11: Comparison between RTAC (real-time) and SAC in $E$ (turn-based). Mean and $95\%$ confidence interval are computed over eight training runs per environment.

Figure 12: RTAC with and without output normalization. Mean and $95\%$ confidence interval are computed over eight and four training runs per environment, respectively.

# B  Hyperparameters

Table 1: Hyperparameters

| Name | RTAC | SAC |
|---|---|---|
| optimizer | Adam | Adam (Kingma & Ba, 2014) |
| learning rate | 0.0003 | 0.0003 |
| discount ($\gamma$) | 0.99 | 0.99 |
| hidden layers | 2 | 2 |
| units per layer | 256 | 256 |
| samples per minibatch | 256 | 256 |
| target smoothing coefficient ($\tau$) | 0.005 | 0.005 |
| gradient steps / environment steps | 1 | 1 |
| reward scale | 5 | 5 |
| entropy scale ($\alpha$) | 1 | 1 |
| actor-critic loss factor ($\beta$) | 0.2 | - |
| Pop-Art alpha | 0.0003 | - |
| start training after | 10000 | 10000 steps |

# C  Proofs

**Theorem 1.** [4] *A policy $\boldsymbol{\pi} : A \times \boldsymbol{X} \to \mathbb{R}$ interacting with $RTMDP(E)$ in the conventional, turn-based manner gives rise to the same Markov Reward Process as $\boldsymbol{\pi}$ interacting with $E$ in real-time, i.e.*

$$RTMRP(E, \boldsymbol{\pi}) = TBMRP(RTMDP(E), \boldsymbol{\pi}). \tag{3}$$

*Proof.* For any environment $E = (S, A, \mu, p, r)$, we want to show that the two above MRPs are the same. Per Def. 2 and 4 for $TBMRP(RTMDP(E), \boldsymbol{\pi})$ we have

(1) state space $\qquad\qquad\qquad S \times A,$

(2) initial distribution $\qquad\quad \mu(s)\delta(a - c),$

(3) transition kernel $\qquad\quad \int_A p(s_{t+1}|s_t, a_t)\delta(a_{t+1} - \boldsymbol{a})\, \boldsymbol{\pi}(\boldsymbol{a}|s_t, a_t)\, d\boldsymbol{a},$

(4) state-reward function $\qquad \int_A r(s, a)\, \boldsymbol{\pi}(\boldsymbol{a}|s_t, a_t)\, d\boldsymbol{a}.$

The transition kernel, using the definition of the Dirac delta function $\delta$, can be simplified to

$$p(s_{t+1}|s_t, a_t) \int_A \delta(a_{t+1} - \boldsymbol{a})\, \boldsymbol{\pi}(\boldsymbol{a}|s_t, a_t)\, d\boldsymbol{a} = p(s_{t+1}|s_t, a_t)\, \boldsymbol{\pi}(a_{t+1}|s_t, a_t). \tag{15}$$

The state-reward function can be simplified to

$$r(s_t, a_t) \int_A \pi(\boldsymbol{a}|\boldsymbol{x})\, d\boldsymbol{a} = r(s_t, a_t). \tag{16}$$

It should now be easy to see how the elements above match $RTMRP(E, \boldsymbol{\pi})$, Def. 3. $\qquad\qquad \square$

**Theorem 2.** *A policy $\boldsymbol{\pi}(\boldsymbol{a}|s, b, a) = \pi(\boldsymbol{a}|s)$ interacting with TBMDP(E) in real time, gives rise to a Markov Reward Process that contains (Def. 10) the MRP resulting from $\pi$ interacting with E in the conventional, turn-based manner, i.e.*

$$TBMRP(E, \pi) \propto RTMRP(TBMDP(E), \boldsymbol{\pi}) \tag{4}$$

*Proof.* Given MDP $E = (S, A, \mu, p, r)$, we have $\Psi = (Z, \nu, \sigma, \bar{\rho}) = RTMRP(TBMDP(E), \boldsymbol{\pi})$ with

(1) state space
$$Z = S \times \{0, 1\} \times A, \tag{17}$$

(2) initial distribution
$$\nu(s, b, a) = \mu(s)\,\delta(b)\,\delta(a - c), \tag{18}$$

(3) transition kernel
$$\sigma(s_{t+1}, b_{t+1}, a_{t+1}|s_t, b_t, a_t) \tag{19}$$

$$= \begin{cases} \delta(s_{t+1} - s_t)\,\delta(b_{t+1} - 1)\,\pi(a_{t+1}|s_t) & \text{if } b_t = 0 \\ p(s_{t+1}|s_t, a_t)\,\delta(b_{t+1})\,\pi(a_{t+1}|s_t) & \text{if } b_t = 1 \end{cases}, \tag{20}$$

(4) state-reward function
$$\bar{\rho}(s, b, a) = r(s, a)\,b. \tag{21}$$

We can construct $\Omega = (Z, \nu, \boldsymbol{\kappa}, \bar{\boldsymbol{r}})$, a sub-MRP with interval $n = 2$. Since we always skip the step in which $b = 1$, we only have to define the transition kernel for $b_t = 0$, i.e.

$$\boldsymbol{\kappa}(z_{t+1}|z_t) = \sigma^2(s_{t+1}, b_{t+1}, a_{t+1}|s_t, b_t, a_t) \tag{22}$$

$$= \int_{S \times A} \sigma(s_{t+1}, b_{t+1}, a_{t+1}|s', 1, a')\,\sigma(s', 1, a'|s_t, 0, a_t)\,d(s', a') \tag{23}$$

$$= \int_{S \times A} p(s_{t+1}|s', a')\,\delta(b_{t+1})\,\pi(a_{t+1}|s')\,\delta(s' - s_t)\,\pi(a'|s_t)\,d(s', a') \tag{24}$$

$$= \int_A p(s_{t+1}|s_t, a')\,\delta(b_{t+1})\,\pi(a'|s_t)\,da'. \tag{25}$$

For the state-reward function we have (again only considering $b = 0$)

$$\bar{\boldsymbol{r}}(s, b, a) = v_\Psi^2(s, b, a) \tag{26}$$

$$= \underbrace{\bar{\rho}(s, 0, a)}_{=0} + \int_{S \times A} \bar{\rho}(s', 1, a')\,\sigma(s', 1, a'|s, 0, a)\,d(s', a') \tag{27}$$

$$= \int_{S \times A} r(s', a')\,\delta(s' - s)\,\pi(a'|s)\,d(s', a') \tag{28}$$

$$= \int_A r(s, a')\,\pi(a'|s)\,da'. \tag{29}$$

The sub-MRP $\Omega$ is already very similar to $TBMRP(E, \pi)$ except for having a larger state-space. To get rid of the $b$ and $a$ state components, we reduce $\Omega$ with a state transformation $f(s, b, a) = s$. The reduced MRP has

(1) state space
$$\{f(z) : z \in Z\} = S, \tag{30}$$

(2) initial distribution
$$\int_{f^{-1}(s)} \nu(z)dz = \int_{\{s\} \times \{0,1\} \times A} \mu(s)\delta(b)\delta(a - c)\,d(s, b, a) = \mu(s), \tag{31}$$

(3) transition kernel
$$\int_{f^{-1}(s_{t+1})} \boldsymbol{\kappa}(z'|z)\,dz' \text{ for almost all } z \in f^{-1}(s_t) \tag{32}$$

$$= \int_{\{s_{t+1}\} \times \{0,1\} \times A} \boldsymbol{\kappa}(z'|z)\,dz' \text{ for almost all } z \in \{s_t\} \times \{0, 1\} \times A \tag{33}$$

$$= \int_A p(s_{t+1}|s_t, a')\,\pi(a'|s_t)\,da' \tag{34}$$

(4) state-reward function
$$\bar{\boldsymbol{r}}(z) \text{ for almost all } z \in f^{-1}(s). \tag{35}$$

$$= \int_A r(s, a')\,\pi(a'|s)\,da', \tag{36}$$

which is exactly $TBMRP(E, \pi)$. $\qquad\square$

**Lemma 1.** *In a Real-Time Markov Decision Process for the action-value function we have*

$$q^{\boldsymbol{\pi}}_{RTMDP(E)}(s_t,a_t,\boldsymbol{a}_t)=r(s_t,a_t)+\mathbb{E}_{s_{t+1}\sim p(\cdot|s_t,a_t)}[\mathbb{E}_{\boldsymbol{a}_{t+1}\sim\boldsymbol{\pi}(\cdot|s_{t+1},\boldsymbol{a}_t)}[q^{\boldsymbol{\pi}}_{RTMDP(E)}(s_{t+1},\boldsymbol{a}_t,\boldsymbol{a}_{t+1})]] \quad (6)$$

*Proof.* After starting with the definition of the action-value function for an environment $(\boldsymbol{X},A,\boldsymbol{\mu},\boldsymbol{p},\boldsymbol{r}) = RTMDP(E)$ with $E = (S,A,\mu,p,r)$, we separate the transition distribution $\boldsymbol{p}$ into its two constituents $p$ and $\delta$ and then, integrate over the Dirac delta.

$$q^{\boldsymbol{\pi}}_{RTMDP(E)}(\boldsymbol{x}_t,\boldsymbol{a}_t) = q^{\boldsymbol{\pi}}_{RTMDP(E)}(s_t,a_t,\boldsymbol{a}_t) \tag{37}$$

$$= \boldsymbol{r}(s_t,a_t,\boldsymbol{a}_t)+\mathbb{E}_{s_{t+1},a_{t+1}\sim\boldsymbol{p}(\cdot|s_t,a_t,\boldsymbol{a}_t)}[\underbrace{\mathbb{E}_{\boldsymbol{a}_{t+1}\sim\boldsymbol{\pi}(\cdot|s_{t+1},a_{t+1})}[q^{\boldsymbol{\pi}}_{RTMDP(E)}(s_{t+1},a_{t+1},\boldsymbol{a}_{t+1})]}] \tag{38}$$

$$= r(s_t,a_t) \quad + \quad \int_S p(s_{t+1}|s_t,a_t) \quad \int_A \delta(a_{t+1}-\boldsymbol{a}_t) \qquad \dots \qquad da_{t+1}\ ds_{t+1} \tag{39}$$

$$= r(s_t,a_t)+\int_S p(s_{t+1}|s_t,a_t)\ \mathbb{E}_{\boldsymbol{a}_{t+1}\sim\boldsymbol{\pi}(\cdot|s_{t+1},\boldsymbol{a}_t)}[q^{\boldsymbol{\pi}}_{RTMDP(E)}(s_{t+1},\boldsymbol{a}_t,\boldsymbol{a}_{t+1})]\ ds_{t+1} \tag{40}$$

$$\square$$

**Lemma 2.** *In a Real-Time Markov Decision Process for the state-value function we have*

$$v^{\boldsymbol{\pi}}_{RTMDP(E)}(s_t, a_t) = r(s_t, a_t) + \mathbb{E}_{s_{t+1}\sim p(\cdot|s_t,a_t)}[\mathbb{E}_{\boldsymbol{a}_t\sim\boldsymbol{\pi}(\cdot|s_t, a_t)}[v^{\boldsymbol{\pi}}_{RTMDP(E)}(s_{t+1}, \boldsymbol{a}_t)]]. \tag{8}$$

*Proof.* We follow the same procedure as for Lemma 1.

$$v^{\boldsymbol{\pi}}_{RTMDP(E)}(\boldsymbol{x}_t) = v^{\boldsymbol{\pi}}_{RTMDP(E)}(s_t,a_t) \tag{41}$$

$$= \mathbb{E}_{\boldsymbol{a}_t\sim\boldsymbol{\pi}(\cdot|s_t,a_t)}[\boldsymbol{r}(s_t,a_t,\boldsymbol{a}_t)+\mathbb{E}_{s_{t+1},a_{t+1}\sim\boldsymbol{p}(\cdot|s_t,a_t,\boldsymbol{a}_t)}[v^{\boldsymbol{\pi}}_{RTMDP(E)}(s_{t+1},a_{t+1})]] \tag{42}$$

$$= r(s_t,a_t)+\mathbb{E}_{\boldsymbol{a}_t\sim\boldsymbol{\pi}(\cdot|s_t,a_t)}[\int_S p(s_{t+1}|s_t,a_t)\int_A \delta(a_{t+1}-\boldsymbol{a}_t)\ v^{\boldsymbol{\pi}}_{RTMDP(E)}(s_{t+1},a_{t+1})\ da_{t+1}\ ds_{t+1}] \tag{43}$$

$$= r(s_t,a_t)+\int_S p(s_{t+1}|s_t,a_t)\ \mathbb{E}_{\boldsymbol{a}_t\sim\boldsymbol{\pi}(\cdot|s_t,a_t)}[v^{\boldsymbol{\pi}}_{RTMDP(E)}(s_{t+1},\boldsymbol{a}_t)]\ ds_{t+1} \tag{44}$$

$$\square$$

**Proposition 1.** *The following policy loss based on the state-value function*

$$L^{RTAC}_{RTMDP(E),\boldsymbol{\pi}} = \mathbb{E}_{(s_t,a_t)\sim D}\mathbb{E}_{s_{t+1}\sim p(\cdot|s_t,a_t)}D_{KL}(\boldsymbol{\pi}(\cdot|s_t,a_t)||\exp(\tfrac{1}{\alpha}\gamma\boldsymbol{v}(s_{t+1},\cdot))/Z(s_{t+1})) \tag{10}$$

*has the same policy gradient as $L^{SAC}_{RTMDP(E),\boldsymbol{\pi}}$, i.e.*

$$\nabla_{\boldsymbol{\pi}} L^{RTAC}_{RTMDP(E),\boldsymbol{\pi}} = \nabla_{\boldsymbol{\pi}} L^{SAC}_{RTMDP(E),\boldsymbol{\pi}} \tag{11}$$

*Proof.* As shown in Haarnoja et al. (2018a), Equation 9 can be reparameterized to obtain the policy gradient, which, applied in a RTMDP, yields

$$\nabla_{\boldsymbol{\pi}} L^{SAC}_{RTMDP(E),\boldsymbol{\pi}} = \mathbb{E}_{\boldsymbol{x}_t,\epsilon}[\nabla_{\boldsymbol{\pi}}(\log\boldsymbol{\pi}(\boldsymbol{h}_{\boldsymbol{\pi}}(\boldsymbol{x}_t,\epsilon),\boldsymbol{x}_t) - \tfrac{1}{\alpha}\nabla_{\boldsymbol{\pi}} q(\boldsymbol{x}_t,\boldsymbol{h}_{\boldsymbol{\pi}}(\boldsymbol{x}_t,\epsilon))] \tag{45}$$

and reparameterizing Equation 10 yields

$$\nabla_{\boldsymbol{\pi}} L^{RTAC}_{RTMDP(E),\boldsymbol{\pi}} = \mathbb{E}_{\boldsymbol{x}_t,\epsilon}[\nabla_{\boldsymbol{\pi}}(\log\boldsymbol{\pi}(\boldsymbol{h}_{\boldsymbol{\pi}}(\boldsymbol{x}_t,\epsilon),\boldsymbol{x}_t) - \tfrac{1}{\alpha}\gamma\nabla_{\boldsymbol{\pi}}\mathbb{E}_{s_{t+1}\sim p(\cdot|\boldsymbol{x}_t)}[\boldsymbol{v}(s_{t+1},\boldsymbol{h}_{\boldsymbol{\pi}}(\boldsymbol{x}_t,\epsilon))]] \tag{46}$$

where $\boldsymbol{h}_{\boldsymbol{\pi}}$ is a function mapping from state and noise to an action distributed according to $\boldsymbol{\pi}$. This leaves us to show that

$$\nabla_{\boldsymbol{a}_t} q(\boldsymbol{x}_t,\boldsymbol{a}_t) = \underbrace{\nabla_{\boldsymbol{a}_t}\boldsymbol{r}(\boldsymbol{x}_t,a_t)}_{=0} + \nabla_{\boldsymbol{a}_t}\gamma\mathbb{E}_{\boldsymbol{x}_{t+1}\sim\boldsymbol{p}(\cdot|\boldsymbol{x}_t,\boldsymbol{a}_t)}[\boldsymbol{v}(\boldsymbol{x}_{t+1})] = \gamma\nabla_{\boldsymbol{a}_t}\mathbb{E}_{s_{t+1}\sim p(\cdot|\boldsymbol{x}_t)}[\boldsymbol{v}(s_{t+1},\boldsymbol{a}_t)] \tag{47}$$

which follows from the definition of the soft action-value function and simplifying quantities defined in the RTMDP. $\square$

# D Definitions

**Definition 7.** *A Turn-Based Markov Decision Process* $(Z, A, \nu, q, \rho) = TBMDP(E)$ *augments another Markov Decision Process* $E = (S, A, \mu, p, r)$, *such that*

*(1) state space* $\qquad\qquad\qquad Z = S \times \{0, 1\}$,

*(2) action space* $\qquad\qquad\qquad A$,

*(3) initial state distribution* $\qquad \nu(s_0, b_0) = \mu(s_0)\, \delta(b_0)$,

*(4) transition distribution* $\qquad q(s_{t+1}, b_{t+1} | s_t, b_t, a_t) = \begin{cases} \delta(s_{t+1} - s_t)\, \delta(b_{t+1} - 1) & \text{if } b_t = 0 \\ p(s_{t+1} | s_t, a_t)\, \delta(b_{t+1}) & \text{if } b_t = 1 \end{cases}$

*(5) reward function* $\qquad\qquad\quad \rho(s, b, a) = r(s, a)\, b$.

**Definition 8.** $\Omega = (Z, \nu, \boldsymbol{\kappa}, \bar{\boldsymbol{r}})$ *is a sub-MRP of* $\Psi = (Z, \nu, \sigma, \bar{\rho})$ *if its states are sub-sampled with interval* $n \in \mathbb{N}$ *and rewards are summed over each interval, i.e. for almost all* $z$

$$\boldsymbol{\kappa}(z'|z) = \kappa^n(z'|z) \quad \text{and} \quad \bar{\boldsymbol{r}}(z) = v_\Psi^n(z). \tag{48}$$

**Definition 9.** *A MRP* $\Omega = (S, \mu, \kappa, \bar{r})$ *is a reduction of* $\boldsymbol{\Omega} = (Z, \nu, \boldsymbol{\kappa}, \bar{\boldsymbol{r}})$ *if there is a state transformation* $f : \boldsymbol{Z} \to S$ *that neither affects the evolution of states nor the rewards, i.e.*

*(1) state space* $\qquad\qquad\qquad S = \{f(z) : z \in Z\}$, $\qquad\qquad\qquad\qquad\qquad$ (49)

*(2) initial distribution* $\qquad\quad \mu(s) = \displaystyle\int_{f^{-1}(s)} \nu(z) dz$, $\qquad\qquad\qquad\qquad$ (50)

*(3) transition kernel* $\qquad\quad \kappa(s_{t+1}|s) = \displaystyle\int_{f^{-1}(s_{t+1})} \boldsymbol{\kappa}(z'|z)\, dz' \ \text{ for almost all } z \in f^{-1}(s),$ (51)

*(4) state-reward function* $\qquad r(s) = \bar{\boldsymbol{r}}(z) \ \text{ for almost all } z \in f^{-1}(s).$ (52)

**Definition 10.** *A MRP* $\Psi$ *contains another MRP* $\Omega$ *(we write* $\Omega \propto \Psi$*) if* $\Psi$ *works at a higher frequency and has a richer state than* $\Psi$ *but behaves otherwise identically. More precisely,*

$$\Omega \propto \Psi \iff \Omega \text{ is a reduction (Def. 9) of a sub-MRP (Def. 8) of } \Psi. \tag{53}$$

**Definition 11.** *The $n$-step transition function of a MRP* $\Omega = (S, \mu, \kappa, \bar{r})$ *is*

$$\kappa^n(s_{t+n}|s_t) = \int_S \kappa(s_{t+n}|s_{t+n-1}) \kappa^{n-1}(s_{t+n-1}|s_t)\, ds_{t+n-1}. \quad \big| \text{ with } \kappa^1 = \kappa \tag{54}$$

**Definition 12.** *The $n$-step value function* $v_\Omega^n$ *of a MRP* $\Omega = (S, \mu, \kappa, \bar{r})$ *is*

$$v_\Omega^n(s_t) = \bar{r}(s_t) + \int_S \kappa(s_{t+1}|s_t) v_\Omega^{n-1}(s_{t+1})\, ds_{t+1}. \quad \big| \text{ with } v_\Omega^1 = \bar{r} \tag{55}$$

## Footnotes

[4]All proofs are in Appendix C.