[Reviews · NeurIPS 2019]

Reviewer 1



Positive: - Overall, I feel that the paper provides an interesting contribution that may help to work toward applying RL to real-world problems where an agent interacts with the physical world, e.g. in robots. + reproducibility: The authors promised in the authors response to make their code available. Negative: - One problem I see with the paper is that it is unclear at this point whether this line of work is necessary because with increased computing power on embedded devices such as robots, the inference time of most methods turns out to actually be neglible (millisecond range or faster). I feel that this point might be alleviated by providing a series of experiments (e.g. in the driving experiment proposed in the paper) where the agent is assumed to be super fast, very fast, fast, not fast, really slow - and show how that impacts the performance of the SAC method. - another problem with the paper is that it is partially hard to follow the notation and reasoning (see more details below). -> which the authors have promised to improve in their response. more detailed comments: line 29: it is hard to udnerstand what the authors mean with "one time-step" here - it becomes clear later that the authors refer to one agent/environemtn step here - but this could also be read as "the agent is really, really fast" -> this is not a big problem though because it becomes clear later (and the figures next to the text also help). Maybe just referring to the figure inline here would already address make this much clearer and prepare the reader better for the rest of the paper. sec 3, first paragraph: the authors start using u for the action and it is difficult to follow here why a new symbol is used there. Maybe stick with a? lines 69ff: - t_\pi is not defined (and I read it as the time it takes to evlauate the policy. - t_s is not defined (and I don't actually see how it is different from t_pi - maybe just use a single symbol here (there is a bit of discussion about choosing t_s to be larger or smaller than t_pi - but I don't see the point in that sections 3.1 / 3.2: This is quite confusing: Section 3.1 defines RTMRP(E) as an environment that behaves the same with turn-based interaction as E would behave with real-time interaction. Section 3.2 defines TB(E) as ? - and RTMRP(TB(E)) as ? It feels like this shoudl be TB(RTMRP(E))? Overall, I do understand that these sections are describing augmentations/reductions that convert real-time RL environments into turn-based environments and vice versa but the description and the notation are quite confusing to me. Maybe it would be easier to follow if: E_{rt} is a real time environment Figure 3/4: could be merged which would make it easier to compare the performance of the "working" SAC with the RT methods.

Reviewer 2



This paper constructs a framework for performing reinforcement learning where while choosing an action, the environment state can simultaneously change. This framework is inspired by real-world systems where the environment runs asynchronously from the agents and we would like agents to learn and act in "real-time". This is an important topic of study for practical application of RL to physical systems, including robotics. The authors show that the framework can represent existing MDPs. They also introduce a new learning method based on the Soft-Actor Critic for learning in this framework. Empirical results show that the real-time algorithm outperforms the existing SAC algorithm on various continuous control tasks. Would this approach handle real-world effects like jitter? I can see a scenario where jitter shifts multiple state changes into the same transition. It seems like your formulation only uses the latest state, so if jitter could cause you to miss assigning rewards for important states. Would effects like this be considered a partially observable RTMDP? Or would you just have to set an action rate as low as the lowest expected delta between state observations due to jitter? Line 78: Is this actually true? I could imagine a scenario where if the agent had taken a lot of small fast actions with t_s < t_\pi it would achieve the desired behavior. But if it waits to act at a slower rate by continuing computation as suggested here, the required action may be outside of the action space of the agent (assuming that its action space is bounded). Making the code available would make this a stronger, more reproducible paper. Similarly for the car simulation used in some of the experiments. Overall, this is an interesting area to study with potentially a lot of impact on real-world RL applications. The experiments show that even in environments where the standard MDP model works well, by considering the problem in the real-time framework, there is potential for even faster learning and better final policies.

Reviewer 3



This paper introduces a new RL framework that considers the time of taking action. The relationship between the new method and classical MDP is also specified in the paper. Based on the new framework, the authors propose a new actor-critic algorithm. The idea of this paper is interesting, and the paper is overall well written. My main concern is that the definition of the real-time RL problem seems to be inconsistent with the motivation of the paper. I understand the authors want to take the time of the action into consideration, but the definition of real-time RL, we cannot see this accurately.

[Author Response · NeurIPS 2019]

We thank the reviewers for their thorough reviews, valuable feedback and great questions. Multiple reviewers suggested
as an improvement to release the code for reproducibility. We intend to release the code for RTAC, our version of SAC,
as well as all the code necessary to reproduce the MuJoCo and car simulation experiments.

We realize that we might not have convincingly communicated the usefulness of the real-time framework beyond
modelling action selection time. We try to address this in our answer to Reviewer 1. However, before that, we want to
clarify the central assumption of the paper, since the relevant paragraph from line 69 ff. seemed to have been quite
confusing. There are two different time spans:

| | | |
|---|---|---|
| timestep size | $t_s$ | (the time between two observations) |
| action selection time | $t_\pi$ | (e.g. time required for a forward pass through the policy network) |

RTRL deals with the special case in which $t_s = t_\pi$. In that case an action $a_t$ does not affect the next state $s_{t+1}$, which
opens up a number of new algorithmic possibilities. We think $t_s = t_\pi$ is the right assumption because it leads to
*back-to-back action selection*. That is, immediately upon finishing to compute an action the next observation is sampled.
This should always be the goal, no matter how little time is required to compute an action. It allows the agent to update
its actions the quickest, e.g. if we could compute an action in 1ms we should do so 1000 times per second.

**Review 1**  We agree, that an experiment with different action selection times for SAC would be interesting. It should
be noted though, that RTAC (with $t_\pi = t_s$) even outperforms the most idealized ($t_\pi = 0$) version of SAC (see Fig. 9 in
the Appendix). One might argue that this comparison between RTAC and SAC is irrelevant because they are different
algorithms. However, the only differences between SAC and RTAC are those made possible by using the real-time
framework, which is actually exactly what we think this paper is about. We believe this line of work is valuable because
by assuming $t_\pi = t_s$ the framework becomes simpler rather than more complex (compare *MRP* and *RTMRP*) and it
gives us the opportunity to create better algorithms.

We used the transformation notation to be able to establish equivalences such as $RTMRP(E, \boldsymbol{\pi}) = MRP(RTMDP(E), \boldsymbol{\pi})$.
However, we agree, that the notation is inconsistent. We refer to both the function and to its result as MRP. We plan to
rename the standard, turn-based *MRP*-function to *TBMRP*. The transformation *TB* will be called *TBMDP* indicating
that, like *RTMDP*, it is a function transforming one MDP into another. The two central equivalences, showing that
real-time interactions can be expressed with turn-based interactions and vice versa, would then look like

$$RTMRP(E, \boldsymbol{\pi}) = TBMRP(RTMDP(E), \boldsymbol{\pi}) \quad \text{and} \quad TBMRP(E, \pi) \propto RTMRP(TBMDP(E), \boldsymbol{\pi}).$$

For the right side we will add proper definitions and proof in the appendix. So far, we neglected the *TBMDP* a bit since
the rest of the paper does not depend on it. As suggested, we will rename $u$ to $a$.

**Review 2**  If we understand it correctly, the question about jitter refers to the problem of aliasing in case we are not
able to sample quickly enough the quantities that we want to observe (Nyquist-Shannon sampling theorem). Both
with turn-based and with real-time interaction, we might run into the problem of not being able to further reduce
the timestep size beyond the action selection time of our agent. In that case, information could "hide" in higher,
unobserved frequencies making the observation process non-markov and thus not representable by a MDP. Especially
if our environment is non-markov for other reasons too, one solution might be to transform the observation process
into a markov process by concatenating all past observations. Another, more complex solution, would be to reduce the
policy evaluation time by breaking the policy up into a pipeline of stages (as mentioned in line 79). All stages could
be evaluated in parallel, reducing computation time and allowing us to lower the timestep size which would solve our
aliasing problem. More precisely, we would construct a "pipelining MDP" where an action would contain the outputs
of all the pipeline stages and a state would contain the inputs to all the stages. The transition function of the pipelining
MDP would simply forward the intermediate computations from one stage to the next (in addition to representing the
actual, underlying state transitions).

We only directly compare RTAC with SAC. With other algorithms it would not have been a like-for-like comparison.
However, we will consider adding TD3 (the only other competitive algorithm) to our comparison and codebase.

**Review 3**  We motivate our paper as being one of the first reformulations of RL to take the *time to select an action*
into consideration, not the "time of the action". We introduce and mathematically define the RTMRP framework, an
interaction framework that assumes that the agent takes exactly one timestep to select an action. We furthermore explain
why the assumed time of a single timestep is a reasonable choice and why we believe it does not limit generality. The
RTRL framework stands in contrast to the conventional RL interaction framework, in which the agent selects the action
for the current timestep, and no time is allowed for selecting the action.

We hope that this response helps Reviewer 3 to see how the content of our paper matches its motivation. Since this
was their only negative comment, we would hope that, with this important distinction being underscored, they might
reassess their "reject" vote.

[Meta-Review · NeurIPS 2019]

This paper received two positive and one negative reviews, and the negative one was very short and non-specific, so normally it would be an accept (and so I will ultimately recommend). A weakness of the paper not noted by the reviewers is that the authors are apparently unaware of the related paper by Travnik et al. (see below). The Travnik paper subtracts slightly from the novelty of the current work but adds to the recognition of the importance of the real-time issues. On balance, I don’t think that the existence of this prior work diminishes the case for publication of this paper. (However, I do think the paper should be revised to refine its claim to novelty and to cite the Travnik paper). reference: Travnik, J. B., Mathewson, K. W., Sutton, R. S., & Pilarski, P. M. (2018). Reactive Reinforcement Learning in Asynchronous Environments. Frontiers in Robotics and AI, 5, 79.